# PCMID: Multi-Intent Detection through Supervised Prototypical Contrastive Learning

**Yurun Song\***
UC Irvine
yuruns@uci.edu

**Junchen Zhao\***
UC Irvine
junchez3@uci.edu

**Ian G. Harris**
UC Irvine
harris@ics.uci.edu

**Spencer B. Koehler**
Aktify. Inc
koehlersb747@gmail.com

**Amir Abdullah**
Aktify. Inc
amirali1985@gmail.com

## Abstract

Intent detection is a major task in Natural Language Understanding (NLU) and is the core component of dialogue systems for interpreting users' intentions based on their utterances. Many works have explored detecting intents by assuming that each utterance represents only a single intent. Such systems have achieved very good results; however, intent detection is a far more challenging task in typical real-world scenarios, where each user utterance can be highly complex and express multiple intents. Therefore, in this paper, we propose **PCMID**, a novel **M**ulti-**I**ntent **D**etection framework enabled by **P**rototypical **C**ontrastive Learning under a supervised setting. The **PCMID** model can learn multiple semantic representations of a given user utterance under the context of different intent labels in an optimized semantic space. Our experiments show that **PCMID** achieves the current state-of-the-art performance on both multiple public benchmark datasets and a private real-world dataset for the multi-intent detection task.

## 1 Introduction

Natural Language Understanding (NLU) is a key component of dialogue systems, particularly for the task of intent detection (Tur, 2011) to interpret users' utterances throughout conversations. Intent detection can essentially be regarded as a semantic classification problem. Given an utterance like *"Show me the type of aircraft that Delta uses,"* intent detection can be implemented as a classifier that predicts an intent label for the utterance such as *"Aircraft Type."* This type of intent detection is often called single-intent detection which has been amply explored in previous works. Many of these works adapt joint models to also capture the relationship between intent detection and slot filling (Goo et al., 2018, Li et al., 2018a, E et al.,

---
**\*** Equally contributed.

2019, Qin et al., 2020a). These models currently achieve very promising results.

Despite the promising results on single-intent detection, the assumption that user utterances each represent only a single intent does not always hold in real-world scenarios where an utterance can represent multiple intents (Li et al., 2018b, Rastogi et al., 2019). For example, consider a natural expansion of the previous sample utterance: *"Show me the type of aircraft that Delta uses and what time zone is Denver in?"*. In this case, the classifier should predict multiple intent labels including *Aircraft Type* and *City* instead of just one. In scenarios such as these, failing to consider multiple intent predictions will result in incomplete Natural Language Understanding and limit the performance of a dialogue system.

In order to deal with this issue, Gangadharaiah and Narayanaswamy (2019) explore the joint multi-intent detection and slot filling task using a multi-task framework with a slot-gated mechanism (Goo et al., 2018). This work incorporates multi-intent information within a single context vector but does not integrate fine-grained intent information for token-level slots, resulting in ambiguous relationships between filled slots and intents. Following this work, Qin et al. (2020b) introduce an intent-slot graph interaction layer to model correlations between slots and intents, but the solution does not scale to cases where conjunctions are implicitly defined in utterances.

Most recently, Wu et al. (2021) propose a label-aware BERT attention network, which achieves the state of the art performance on multi-intent detection. This work first applies two BERT encoders, one to capture the semantics of utterances without explicit conjunctions and the other for the words in the intent labels. Then, a label-aware layer is constructed which iteratively projects intent label representations to utterance representations in order to compute a projection weight that represents

the similarity between utterance and intent labels for the purpose of multi-intent detection.

Although this intuitive approach works well for detecting multiple intents, it fails to take the high complexity of the intent label embedding space into account and does not consider the semantics between individual utterance tokens and the intent label words. This results in ambiguity in matching intent labels to utterances such that the related intent information for each utterance token cannot be captured. Furthermore, because two BERT encoders are applied, model size and computational efficiency become issues for both the training and inference phases.

Therefore, in this paper, we propose our framework **PCMID**[1]: Multi-Intent Detection through Supervised Prototypical Contrastive Learning, with the following **main steps and associated contributions**:

- We first obtain the utterance and multi-intent label embedding by constructing an utterance-label encoder using a *single* **BERT encoder**. This achieves a smaller overall model size with higher computational efficiency.

- Next, we obtain the utterance-label attention by passing the utterance and multi-intent label embedding to a multi-head attention block that builds the relationship between each utterance token and its respective multi-intent labels, allowing each utterance token to capture information relevant to its intent labels.

- Finally, we apply multi-label Supervised Prototypical Contrastive Learning to learn multiple semantic representations of a given user utterance under the context of different intent labels in an optimized semantic embedding space of intent labels and utterance-label attention.

We conduct experiments on multiple public benchmark datasets and a private real-world dataset and show that our approach achieves the current state-of-the-art performance on the multi-intent detection task.

## 2    Related Works

### 2.1    Intent Detection

Early works apply traditional machine learning approaches such as Support Vector Machines (Mendoza and Zamora, 2009) and graph based methods (Hu et al., 2009) for intent detection tasks. With the prevalence of deep neural network methods, later works achieve intent detection via RNNs (Ravuri and Stolcke, 2015, Sreelakshmi et al., 2018), CNNs (Gupta et al., 2019; Hashemi et al., 2016), Capsule Networks (Xia et al., 2018) and Transformer-based models (Vaswani et al., 2017; Castellucci et al., 2019; Chen et al., 2019; Zhang et al., 2019; Mehri and Eric, 2021).

Although intent detection works that focus on the single intent use-case have achieved great performance on this task, user utterances are not restricted to conveying one intent at a time in practice. Consequently, multi-intent detection is needed for advancing this research field. The work by (Xu and Sarikaya, 2013) applies a log-linear model atop n-gram features for multi-label detection. Rychalska et al. (2018) propose a hierarchical model for multi-intent detection. Gangadharaiah and Narayanaswamy (2019) propose joint multi-intent detection and slot filling using a multi-tasking framework and slot-gated mechanism but cannot resolve ambiguous relationships between slots and intents. Qin et al. (2020b) propose an adaptive intent-slot graph interaction layer which mitigates the issue of ambiguous relationships from the previous work but requires intents to be detectable by the presence of conjunctions in utterances. Wu et al. (2021) propose a label-aware BERT attention network which is able to detect implicit intents without explicit conjunctions.

### 2.2    Contrastive Learning (CL)

Contrastive Learning (CL) is a specific form of self supervised learning, that focuses on using "positive" and "negative" samples to improve the latent embedding space of models for better classification results. The core idea of CL is finding a strong manifold representation via pulling "positive" pairs closer together, and repelling negative samples under a dot product similarity metric. This idea was first applied in the field of Computer Vision and found great success in multiple tasks such as Data Augmentation (Gao et al., 2021a, Meng et al., 2021, Yan et al., 2021) and Instance Discrimination (Wu et al., 2018, Oord et al., 2018, Ye et al., 2019, He et al., 2019).

Although previous works in CL have achieved great performance, many of them do not consider the semantic structure of data, a fact which usually

---

[1] https://github.com/zjc664656505/PCMID

leads to samples being treated as either positive or negative pairs as long as they are from different instances without considering their semantic similarity. This results in samples with shared semantic similarity being pushed apart in the semantic space and prevents the model from improving. To address this issue, Li et al. (2020) propose Prototypical Contrastive Learning (PCL), which accounts for semantic structure. Khosla et al. (2020) propose Supervised Contrastive Learning which extends the self-supervised CL to fully-supervised CL and more effectively utilizes the label information in CL. Dao et al. (2021) propose multi-label Supervised CL that allows model to learn multiple representations of an image under the context of different given labels.

In this paper, we apply the principles from these prior works to the multi-intent detection task by applying multi-label supervised CL, and PCL. Finally, we show our approach's effectiveness in our experiments.

## 3 Approach

In this section, we describe in detail our proposed model architecture, **PCMID**, as illustrated in Figure 1. The architecture consists of an utterance-label encoder module that uses a *single* **BERT encoder**, a multi-head attention block module, and a multi-label supervised contrastive learning module.

### 3.1 Utterance-Label encoder

The first component in our model architecture is an utterance-label encoder, as shown at the bottom of Figure 1, based on BERT and denoted as $BERT_\psi$ below. This encoder is used to obtain both a token-level embedding of the utterance and a sentence-level embedding of the intent label.

Given the utterance inputs $X = (x_1, x_2, ..., x_n)$, we obtain token-level embeddings for each utterance by simply feeding each $x_i$ into the $BERT_\psi$ encoder. Given the intent label inputs $Y = (l_1, l_2, ..., l_m)$, we also compute the intent label sentence-level embeddings by first feeding each $l_j$ to the $BERT_\psi$ encoder with a pooling mechanism to get the sentence-level hidden representation of each intent label and then stacking them together. Constructing the intent label embedding at the sentence level enables the model to learn the hidden representation of multiple intent labels given a single utterance such that each $x_i \in X$ is associated with multiple intent labels selected from the set of

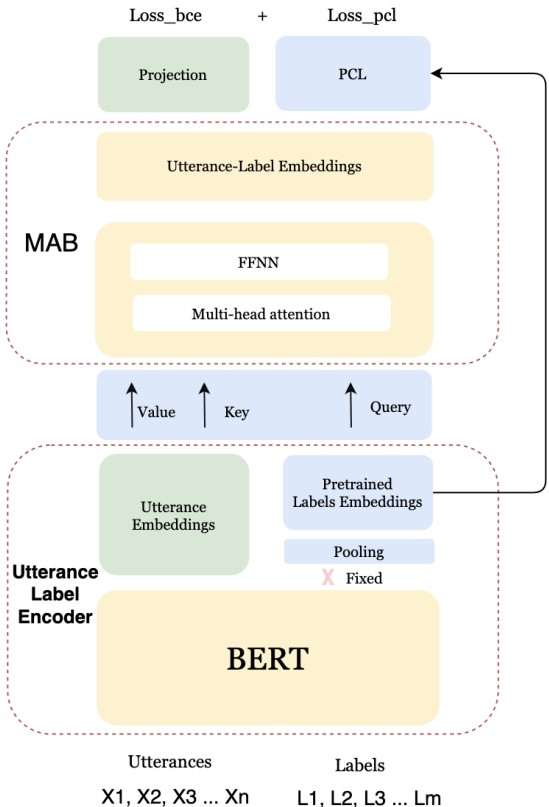

Figure 1: **Model Architecture** PCMID is mainly composed by three components, which are Utterance-Label Encoder, Multi-head Attention Block and Prototypical Contrastive learning.

$M$ labels in total. Formally, the process of obtaining the utterance embedding, $E_X$, and intent label embedding, $E_Y$, is defined as,

$$E_X = BERT_\psi(X)$$
$$E_Y = Stack([h_{y_1}, h_{y_2}, ...h_{y_m}]) \quad (1)$$
$$\textbf{where} \quad h_{y_j} = pooling(BERT_\psi(l_j))$$

In Equation 1, $E_X \in \mathbb{R}^L$ is the token-level embedding of utterance $X$ with a maximum sequence length of $L$ and an embedding size of $H$. $E_Y \in \mathbb{R}^{M \times H}$ is the intent label sentence-level embedding with the label input size of $M$. $h_{y_j} \in \mathbb{R}^{1 \times H}$ is the sentence-level hidden representation of each label $l_j$, where $j \in \{1, \dots m\}$, constructed using pooling methods including default BERT pooling, mean pooling and self attentive pooling Wu et al. (2021) to convert the token-level intent label hidden representations to a sentence-level intent label hidden representation.

### 3.2 Multi-head Attention Block

The next component of **PCMID** is the Multi-head Attention Block (MAB) inspired by the work

of Dao et al. (2021) shown in the middle of Figure 1. The Multi-head Attention Block is adapted from the BERT encoder without considering the positional embedding.

**Multi-head Attention.** Multi-head attention is an extension of attention which is first introduced by Vaswani et al. (2017). Compared to single attention, multi-head attention projects Query, Key, Value to $h$ different head vectors separately. Then, the attention is applied to the $h$ projections individually and results in a linear transformation of the combination of all attention outputs. By applying multi-head attention, the model can attend to information from different hidden representation subspaces at different positions. In our work, the Query is the intent label embedding $E_Y$, the Key and Value are the utterance embedding $E_X$. We express multi-head attention as follows:

$$\text{MHA}(E_Y, E_X, E_X)) = \text{Concat}(O_1, ...O_h)W^o$$
$$\textbf{where } O_h = \text{Att}(E_Y W^q_{y_h}, E_X W^k_{x_h}, E_X W^v_{x_h}) \tag{2}$$

where the $W^o, W^q_{y_h}, W^k_{x_h}, W^v_{x_h}$ are the parameters which are learnable and $E_Y W^q_{y_h}, E_X W^k_{x_h}, E_X W^v_{x_h}$ are the projected vectors based on our Query, Key and Values.

**Multi-head Attention Block.** Based on multi-head attention, we define the Multi-head Attention Block (MAB) which is adapted from the encoder block of BERT, without considering the positional embedding and dropout. In our work, the MAB can be viewed as a 12-layer BERT model with an extra layer on top of the BERT model, taking $E_Y$ as the Query and $E_X$ as the Key and Value. By designing the MAB in this way, each intent label queries the utterance Keys and determines whether the utterance is close to the intent label semantically. The MAB also obtains an utterance-label embedding, denoted as $G$, which represents the semantic association between each utterance's hidden representation in both its embedding and its intent label embedding. The utterance-label embedding $G$ through MAB is formally defined such that:

$$G = \text{MAB}(E_Y, E_X, E_X)$$
$$= \text{LayerNorm}(E'_Y + \text{FFNN}(E'_Y)) \tag{3}$$
$$E'_Y = \text{LayerNorm}(E_Y + \text{MHA}(E_Y, E_X, E_X))$$

Then, we have the output $G \in \mathbb{R}^{N \times M \times H}$ where $N$ is the batch size of the utterance samples, $M$ is the number of intent labels, and $H$ is the embedding size.

### 3.3 Multi-label Supervised Contrastive Learning

The final component of our work is the Multi-label Supervised Contrastive Learning module, which is done through Contrastive Learning (CL) and Prototypical Contrastive Learning (PCL), as shown at the top of Figure 1 and as described in the following subsections.

**Supervised CL.** Contrastive Learning (CL) has been increasingly popular and has proven to be an effective learning mechanism. It shows extraordinary performance for optimizing sentence semantic hidden representations in an embedding space in a single-label setting and in a self-supervised way (Gao et al., 2021b; Chen et al., 2022; Ke et al., 2021). CL helps improve the performance of models, but, as a self-supervised technique, CL does not leverage the label information and the model's learning capability is limited thereby. Consequently, Khosla et al. (2020) propose Supervised CL (SCL) to utilize the label information by selecting positive and negative samples to achieve a significant improvement in model performance. Such improvements are also realized in similar works, like (Liu et al., 2021; Hu et al., 2022; Gunel et al., 2020). Motivated by these works, we adapt SCL to the multi-intent detection task.

Naively extending SCL to multi-intent detection can be complicated by a label mismatch issue. For multi-label SCL, if we have multiple positive samples with an unequal number of negative samples, SCL is apt to find erroneous manifold representations, which in turn would decrease the model performance.

In order to solve this problem, we implement SCL in a multi-label setting by applying our constructed utterance-label embedding $G$. In this embedding, since each utterance's hidden representation is correlated with each intent label's hidden representation in their respective embeddings, the number of positive samples and negative samples will be the same and the label mismatch issue is avoided.

Given the utterance-label embedding $G$ with $N$ utterance instances in each mini-batch, we represent each utterance's hidden representation $i$ with respect to intent label hidden representation

$j$ such that $\{h(i,j) \in G(i,j)|i \in \{1, \ldots N\}; j \in \{1, \ldots M\}\}$ and we represent the ground truth of each utterance sample $i$ regarding each intent label $j$ such that $\{y_{ij} \in \{0,1\}|i \in \{1,2,3,...,N\}; j \in \{1,2,3,...,M\}\}$. Hence, each multi-intent utterance $i$ can be transformed to the single-intent representation $h(i,j)$ with respect to a single ground truth label; so, the multi-intent problem can be solved as single-intent classification. For SCL, we regard $h_{ij}$ as an anchor with the goal of pulling the $h_{ij}$ with the same intent label $j$ closer. Accordingly, we define the set containing all positive samples of $h_{ij}$ as $P(i,j) = \{h_{zj} \in G(i,j)|y_{zj} = y_{ij} = 1\}$ and the set containing all negative samples as $S(i,j) = G \setminus P(i,j)$. Then, we define the SCL loss as:

$$\ell_{scl} = \frac{-1}{|P(i,j)|} \sum_{h_p \in P(i,j)} log \frac{exp(h_{ij} \cdot h_p / \tau)}{\sum_{h_s \in S(i,j)} exp(h_{ij} \cdot h_s / \tau)} \tag{4}$$

where $h_p \in P(i,j)$ is the positive sample, $h_s \in S(i,j)$ is the negative sample and $\tau$ is its temperature.

**Supervised Prototypical Contrastive Learning** Li et al. (2020) propose self-supervised Prototypical Contrastive Learning (PCL), which adapts a clustering mechanism to CL in image tasks. Works including (Medina et al., 2020; Yue et al., 2021) use PCL for the pre-training of a few-shot model and achieve great performance. Wang et al. (2021) apply PCL for improving the label semantic embedding space for the zero-shot slot-filling task. Unlike self-supervised PCL, which finds the distribution of prototypes via clustering, we adapt PCL as a supervised method which marks the labels as prototypes and optimizes the representation of both the intent labels and the utterance samples in the utterance-label embedding space $G$.

The definition of the positive set of samples and negative set of samples differs between PCL and CL. In PCL, we set the prototype to the intent label embedding $E_Y \in \mathbb{R}^{M \times H}$. Then, we define $Q(i,j)$ as the hard positive set of samples which satisfies that $\{h_q(i,j) \in E_Y|y_{ij} = 1\}$ and $C(i,j)$ is a hard negative set of samples which satisfies $\{h_c(i,j) \in E_Y|y_{ij} \neq 1\}$. We define the Supervised Prototypical Contrastive Learning (SPCL) loss as:

$$\ell_{spcl} = \frac{-1}{|Q(i,j)|} \sum_{h_q \in Q(i,j)} log \frac{exp(h_{ij} \cdot h_q / \tau)}{\sum_{h_c \in C(i,j)} exp(h_{ij} \cdot h_c / \tau)} \tag{5}$$

We combine the multi-label classification loss, the Binary Cross-Entropy loss (BCE), and the Supervised PCL loss together during training, for our work's (PCMID's) loss as:

$$\ell_{PCMID} = \ell_{BCE} + \gamma \ell_{spcl} \tag{6}$$

where the parameter $\gamma$ is used to control the trade-off between the two losses.

## 4 Experiment

**Datasets.** We conduct experiments on three popular public datasets including MixATIS[2], MixSNIPS[3] and Facebook Semantic Parsing System (FSPS)[4], which contain 18, 7, and 25 unique intent labels, and 14749, 44173, and 44783 utterance samples, respectively (Qin et al., 2020b, Coucke et al., 2018, Gupta et al., 2018). We also use a real-world private commercial dataset in the credit repair domain, referred to as CREDIT16[5], which contains 16 unique intent labels and 4384 utterance samples. This dataset reflects the real-world challenges regarding the multi-intent detection task.

We pre-process the MixATIS, MixSNIPS and FSPS datasets following the data processing pipeline from the previous work (Wu et al., 2021; Qin et al., 2020b). Also, we pre-process the CREDIT16 dataset by lower-casing and then removing punctuation characters and emojis from the utterances and intent labels. Next, we remove new-line characters and repeated white space characters from each utterance sample. We split the CREDIT16 dataset into train, validation and test sets using the ratio of [0.8, 0.1, 0.1], respectively. The detailed dataset characteristics are shown in Table 1.

| | MixATIS | MixSNIPS | FSPS | CREDIT16 |
|---|---|---|---|---|
| Num. Label | 18 | 7 | 25 | 16 |
| Num. Utt | 14749 | 44173 | 44783 | 4384 |
| Train | 13162 | 397768 | 31279 | 3570 |
| Validation | 828 | 2199 | 9042 | 439 |
| Test | 759 | 2198 | 4462 | 439 |

Table 1: Dataset Characteristics

---

[2]https://github.com/zjc664656505/PCMID/tree/main/data/MixATIS_clean
[3]https://github.com/zjc664656505/PCMID/tree/main/data/MixSNIPS_clean
[4]https://github.com/zjc664656505/PCMID/tree/main/data/SNIPS
[5]https://github.com/zjc664656505/PCMID/tree/main/data/Aktify

| | MixATIS | | MixSNIPS | | FSPS | | CREDIT16 | |
|---|---|---|---|---|---|---|---|---|
| Models | Acc | F1 | Acc | F1 | Acc | F1 | Acc | F1 |
| Stack-Prop | 0.719 | 0.790 | 0.946 | 0.976 | 0.723 | 0.911 | - | - |
| Joint-MID-SF | 0.731 | 0.806 | 0.951 | 0.980 | 0.780 | 0.877 | - | - |
| AGIF | 0.758 | 0.812 | 0.953 | 0.980 | 0.749 | 0.914 | - | - |
| LABAN | 0.765±0.3% | 0.905±0.4% | 0.955±0.3% | 0.980±0.2% | 0.910±0.1% | 0.937±0.1% | 0.447±0.9% | 0.716±0.7% |
| PCMID$_{base}$ pooler | 0.773±0.5% | 0.912±0.2% | 0.960±0.2% | 0.983±0.1% | 0.913±0.1% | 0.941±0.1% | 0.471±1.2% | 0.759±0.8% |
| PCMID$_{base}$ mean | 0.792±0.6% | 0.911±0.7% | 0.959±0.7% | 0.982±0.6% | 0.910±0.1% | 0.940±0.1% | 0.482±1.4% | 0.747±0.7% |
| PCMID$_{large}$ pooler | **0.837*±0.7%** | **0.940*±0.7%** | 0.960±0.2% | **0.983±0.1%** | 0.916±0.1% | 0.942±0.1% | 0.478±0.6% | 0.745±0.6% |
| PCMID$_{large}$ mean | 0.819±1.2% | 0.933±0.8% | 0.960±0.3% | 0.982±0.1% | **0.917*±0.1%** | **0.944*±0.1%** | **0.495*±1.0%** | **0.759±0.4%** |
| S-PCMID$_{base}$ pooler | 0.816±0.9% | 0.917±0.5% | **0.961*±0.4%** | 0.983±0.2% | 0.912±0.2% | 0.940±0.1% | 0.472 ±1.4% | 0.742±0.8% |
| S-PCMID$_{base}$ mean | 0.809±1.5% | 0.919±0.7% | 0.958±0.5% | 0.982±0.1% | 0.913±0.2% | 0.941±0.1% | 0.490±0.7% | 0.749±0.5% |

Table 2: Multi-intent detection results on four datasets. We report accuracy (Acc) for all intent exact matches and F1 scores based on individual intent calculations. * indicates the significant improvement of p-value < 0.05 compared to the previous models. $base$ indicates BERT-base, and $large$ indicates BERT-large.

**Baseline Experiments.** We conduct experiments comparing our approach to previous works including:

***Stack-Prop*** (Qin et al., 2019), which proposes a shared self-attentive encoder and two decoders for joint intent and slot filling tasks separately.

***Joint-MID-SF*** (Gangadharaiah and Narayanaswamy, 2019), which uses Bi-LSTMs with a slot-gated mechanism for slot filling and intent detection tasks.

***AGIF*** (Qin et al., 2020b), which adapts graph BI-LSTM interactive architectures for both slot filling and intent detection tasks simultaneously.

***LABAN*** (Wu et al., 2021), which encodes both utterances and intent labels through two different encoders, along with an adaptive label-aware attentive layer for the multi intent detection task specifically.

**Experimental Setup.** In our work, **PCMID**, we conduct experiments based on the pre-trained BERT-Base-Uncased language model. We apply a pre-trained sentence embedding, the BERT-Base-NLI-STSB-Mean-Tokens[6] from sentence transformer (Reimers and Gurevych, 2019), to further improve our model's semantic understanding capability in our later experiments. Additionally, we have also conducted experiments using pre-trained BERT-Large-Uncased model in our PCIMID framework. We use BERT default pooling (PCMID-pooler), mean pooling (PCMID-mean) and self-attentive pooling (S-PCMID-pooler) in our experiments. Our experiments are conducted using an Nvidia Titan RTX with 24GB GPU memory.

During our training we use the AdamW optimizer as our parameter optimizer, apply StepLR as

---

our learning rate scheduler, in which it adjusts the learning every 4 training steps with a multiplicative factor of learning rate decay equal to 0.9. We set the training epochs, initial learning rate, and batch size hyperparameters to 50, $1e^{-5}$, and 128, respectively in our experiments on the MixATIS, MixSNIPS, and FSPS datasets. However, in our CREDIT16 experiment, we failed to converge within the 50 training epochs, so we increased the number of training epochs to 100 for the CREDIT16 dataset.

We set the threshold for multi-label classification to 0.5 and evaluate the results based on the F1-score and Accuracy metrics. We run the training experiments 10 times per dataset and obtain the trained model parameters based on the best validation accuracy epoch for computing the testing results. We report results of the final model by averaging 5 testing results per dataset.

## 5 Results

**Multi-Intent Results.** Our main experimental results for the multi-intent detection task are shown in Table 2. One of the previous works which we compare to is LABAN (Wu et al., 2021), however we could not reproduce the results presented in their publication. Upon inspection of their github repository, we found that their published results use their validation set as part of their test set. We generated the LABAN results shown in Table 2 by using the code provided in their github repository, but we separated the test set from the validation set.

We find that all PCMID models achieve close results on the MixSNIPS datasets. **S-PCMID$_{base}$ pooler**, which is PCMID with BERT default pooling using pre-trained sentence embeddings, is slightly better than other models. This fact suggests that the model can capture better contex-

---

[6] https://huggingface.co/sentence-transformers/bert-base-nli-stsb-mean-tokens

| | Samples | Labels | Predictions |
|---|---|---|---|
| 1 | give me the fares from miami to cleveland next sunday and which airline is us | atis_airfare, atis_airline | atis_airfare, atis_airline |
| 2 | list la and how much is a limousine service in la guardia | atis_city, atis_ground_fare | atis_ground_fare |
| 3 | understand that i already have safety alert center and it's too expensive for what i | already_client, money_too_expensive | already_client, money_too_expensive |
| 4 | 1 i don't have debt with you 2 i ignore your calls emails and messages because i'm no longer interested | don't_owe_anything | stop_calling, don't_owe_anything |

Table 3: Real samples extracted from the MixATIS and CREDIT16 Datasets via the $PCMID_{base}$ mean model.

| | ATIS | SNIPS |
|---|---|---|
| Models | Acc | Acc |
| Stack-Prop | 0.969 | 0.980 |
| Joint MID-SF | 0.954 | 0.972 |
| AGIF | 0.971 | 0.981 |
| LABAN | 0.976 ±0.2% | 0.977 ± 0.2% |
| $PCMID_{base}$ pooler | 0.968 ±0.6% | 0.979 ± 0.4% |
| $PCMID_{base}$ mean | 0.979 ±0.2% | 0.982 ± 0.2% |
| $PCMID_{large}$ pooler | 0.973 ±0.4% | 0.980 ± 0.2% |
| $PCMID_{large}$ mean | **0.980** ±0.2% | **0.982** ± 0.1% |
| S-$PCMID_{base}$ pooler | 0.978 ±0.3% | 0.980 ± 0.4% |
| S-$PCMID_{base}$ mean | 0.977 ±0.2% | 0.981 ± 0.3% |

Table 4: Single-intent detection results on two datasets. We report the accuracy (Acc) for the verbatims labelled correctly.

tual semantic information by applying the pre-trained sentence embeddings even though the model size is smaller than other $PCMID_{large}$. We also find that our **$PCMID_{large}$ pooler** achieves the highest performance on the MixATIS dataset. **$PCMID_{large}$ mean**, which is PCMID with mean pooling using pre-trained BERT-large word embeddings, achieves the highest accuracy and the best F1 score on both the CREDIT16 and FSPS datasets.

Our work, PCMID, outperforms previous baseline results on every public benchmark dataset with only a single BERT encoder. Regarding our private CREDIT16 dataset, we compare our work with LA-BAN and BERT and show that our work still gets the best performance. However, we do not compare our work with the other baseline models (Stack-Prop, Joint-MID-SF, AGIF) on the CREDIT16 dataset since CREDIT16 is not designed for the joint multi-intent detection and slot filling task, and is therefore not comparable to these works.

**Single-Intent Results.** In order to show the robustness of our work, we also evaluate our approach in the single-intent setting using ATIS[7] and

SNIPS[8] datasets and compare the experiment results of our work with previous works. As can be seen in Table 4, our **$PCMID_{large}$ mean** approach using BERT-large pre-trained word embedding achieves the highest performance on the single-intent detection task for the accuracy evaluation metrics. We observe that although the other results from our approach, such as $PCMID_{base}$ pooler and S-$PCMID_{base}$ pooler, are not achieving accuracy results as good as the $PCMID_{large}$ mean, they still perform relatively well on both datasets and have a minimal performance gap compared to previous works.

## 6 Analysis

**PCL on Multi-Intent Detection.** In Table 5, we present the results of PCMID with different combinations of options between CL and PCL and compare them on all of the datasets we use. Given the results shown in Table 5, we can see that by applying PCL, PCMID outperforms PCMID without any CL strategy on the MixSNIPS, FSPS and CREDIT16 datasets. Especially on the CREDIT16 dataset, using PCL has an absolute advantage over PCMID alone. This indicates that PCL plays an important role in the PCMID model, especially when the amount of training data is limited. When we compare the results of PCMID with CL only and PCMID without CL or PCL, we also see that PCMID's performance is substantially improved on MixATIS and MixSNIPS datasets, but not on the FSPS and CREDIT16 datasets. This suggests that PCMID with CL might be affected by data volume and complexity.

**Intents Visualization** To illustrate the effectiveness of PCL optimization, Figure 2 shows the distribution of label representations of the test samples

main/data/ATIS

[8] https://github.com/zjc664656505/PCMID/tree/main/data/SNIPS

[7] https://github.com/zjc664656505/PCMID/tree/

| Dataset | | MixATIS | | MixSNIPS | | FSPS | | CREDIT16 | |
|---|---|---|---|---|---|---|---|---|---|
| With CL | With PCL | Acc | F1 | Acc | F1 | Acc | F1 | Acc | F1 |
| No | Yes | 0.773 | 0.912 | 0.960 | **0.983** | **0.913** | **0.941** | **0.471** | 0.759 |
| Yes | No | 0.783 | **0.923** | 0.959 | 0.982 | 0.904 | 0.940 | 0.437 | 0.740 |
| Yes | Yes | **0.788** | 0.919 | **0.961** | 0.982 | 0.909 | 0.939 | 0.466 | **0.761** |
| No | No | 0.767 | 0.905 | 0.958 | 0.981 | 0.911 | 0.940 | 0.448 | 0.756 |

Table 5: Analysis of contrastive learning strategies for $PCMID_{base}$. The first row is our default strategy through the $PCMID_{base}$ pooler.

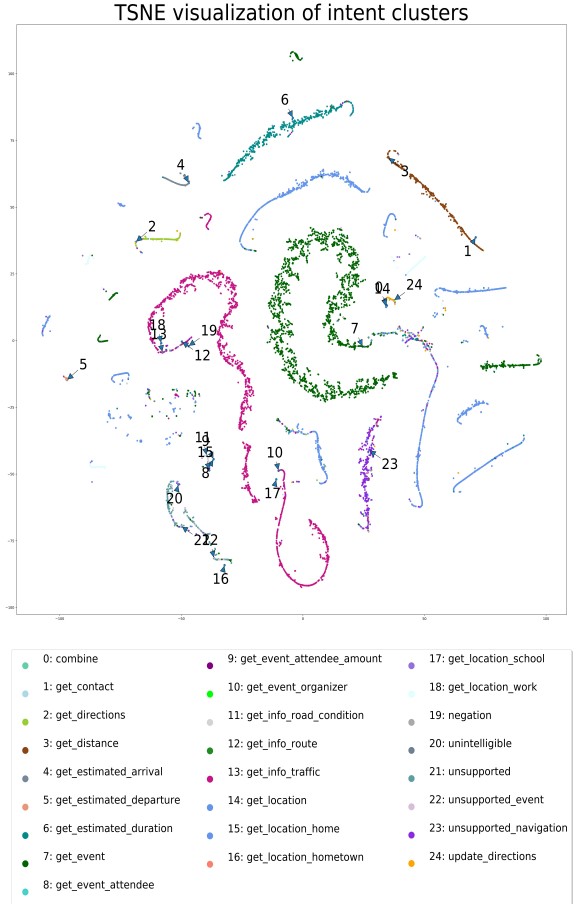

Figure 2: The distribution of FSPS intent representations of test samples from the $PCMID_{base}$ pooler.

from FSPS. Each cluster is clearly separated without significant overlap. Compared with the plot diagram from (Wu et al., 2021), our clusters are more concentrated within the class and more isolated from each other. In other words, due to the PCL optimization, each utterance can be more easily classified into their label clusters.

**Samples Analysis** As a motivational example, consider cases 1 and 2 in the Table 3, from the MixATIS dataset. PCMID is able to predict all labels correctly in case 1. However, it also shows in case 2 that PCMID cannot understand the representation of "la", and misses the label "city" in its prediction.

It indicates that the model can be further trained with some domain specific terms like location, company, etc. Because the CREDIT16 dataset contains real world user conversations, the utterances in the dataset contain some noise. When faced with the imperfect case 3 (incomplete sentence) from the CREDIT16, PCMID is able to make the correct prediction. Even if human labels are corrupted, as in case 4, PCMID makes a correct classification ignoring the noise. In summary, PCMID is a robust model, but it can be further improved by learning the representation relationships between specific terminologies and labels.

In terms of model size, LABAN's (Wu et al., 2021) parameters are about twice that of our PCMID due to LABAN's use of a separate BERT encoder for the label. But their performance is worse than our model. Additionally, BERT fine-tuning classification can yield a decent performance gain on some tasks. By simply building a cross-attention layer of target and utterances on top of BERT, the model can be further boosted following the PCL strategies. With small additional costs, the model can gain consistent improvements, especially in the multi-label tasks.

# 7 Conclusion

In our paper, we propose a multi-label classification model PCMID with a single BERT encoder to handle the multi-intent detection task with smaller model size and higher computational efficiency but still achieve higher performance compared with previous state-of-art work.

Our approach maps complex utterance states to an utterance-label representation with respect to a joint label representation, and adapts prototypical contrastive learning to optimize both the utterance encoder and the representation. Experiments and analysis demonstrate the effectiveness of our proposed model, which can further alleviate the discrepancies between utterance and label representations and improve the quality of label

representations and the performance of multi-label classification. In the future, we will explore how to further optimize the label representations in cross-domains and study the effectiveness of the model on domain transfer tasks.

## Limitations

Although our work achieves good results on the multi-intent detection tasks, we are focusing only on single-domain multi-intent detection and not on cross-domain settings; a fact which limits our work's generalizability.

Through our experiments, we find that the performance of PCMID is sensitive to the initialization of label embeddings. When the label name is not well defined, the pre-trained label embedding through the BERT encoder can reduce the performance, which is even worse than with randomly initialized label embeddings. For example, if the intent labels are not clearly defined, the model will potentially be misled and optimize both utterance embeddings and label embeddings in the wrong direction at the beginning. As a result, utterances will be assigned to the wrong intent label clusters.

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

# Appendix

## Ablation analysis

We focus on the impact of choices in architecture and contrastive learning strategy on the performance of our models, and vary the pooling strategy choice of contrastive learning strategy and label pre-training strategy.

### Self-Attentive vs Mean-Pooling vs Pooler

We experiment with different choices of pooling strategy, as seen in Table 6 namely with mean pooling, a self-attentive layer over the pooled output, and using a pooler layer. We find that at lower accuracy levels, such as seen for MixATIS, mean-pooling is superior to a pooler layer, obtaining a gain of 1.9% in accuracy. Whereas for higher accuracy datasets such as MixSNIPS and FSPS, the pooling strategies are essentially equivalent.

In the Table 6, a comparison is made of the pooling strategies. The performance of three strategies are close for MixSNIPS and FSPS. While using the mean-pooling shows a slight better performance in both MixATIS and CREDIT16 than the other two datasets. Therefore, using the default pooler and mean-pooling is recommended because the use of a self-attentive layer requires a large number of model parameters.

### Label Pre-trained

We validate the effect of label training on our model, with the other loss and pooler strategies kept at the default values. We find that the accuracy is consistently higher when using label pretraining, ranging from a 0.2% accuracy boost on FSPS, to a 1.3% accuracy boost on MixATIS.

In the Table 7, the results show that our models can achieve a better result with pretrained label states in the MixATIS, MixSNIPS and FSPS. But it is not as good as that of using random initialization on the CREDIT16 dataset.

With pretrained label states, the model can handle unseen labels more confidently because unseen labels can be turned into a processed representation instead of being randomly initialized.

However, there are some cases in which the label names are too sparse to represent the utterance's complex intents. In such scenarios, the model can show a better performance without pretrained label states, as seen for CREDIT16 in Table 7.

| Dataset | MixATIS | | MixSNIPS | | FSPS | | CREDIT16 | |
|---|---|---|---|---|---|---|---|---|
| Models | Acc | F1 | Acc | F1 | Acc | F1 | Acc | F1 |
| PCMID$_{base}$ pooler | 0.773 | **0.912** | **0.960** | **0.983** | **0.913** | **0.941** | 0.471 | **0.759** |
| PCMID$_{base}$ mean | **0.792** | 0.911 | 0.959 | 0.982 | 0.910 | 0.940 | **0.482** | 0.747 |
| PCMID$_{base}$ self-attentive | 0.777 | 0.905 | **0.960** | 0.982 | 0.910 | 0.939 | 0.481 | 0.741 |

Table 6: Ablation analysis of pooling strategies for label states.

| Dataset | MixATIS | | MixSNIPS | | FSPS | | CREDIT16 | |
|---|---|---|---|---|---|---|---|---|
| Uses Pretrained Label | Acc | F1 | Acc | F1 | Acc | F1 | Acc | F1 |
| Yes | **0.773** | **0.912** | **0.960** | **0.983** | **0.913** | **0.941** | 0.472 | 0.742 |
| No | 0.760 | 0.899 | 0.957 | 0.981 | 0.911 | 0.940 | **0.486** | **0.747** |

Table 7: Ablation analysis of label pretraining for PCMID. The first row is the default strategy through PCMID-pooler.

## MAB Analysis

In the main experiments, we achieve good results on multi-intent detection tasks across different datasets by setting label embedding as query and utterance embedding as key&value in MAB. In order to ensure this is the best setting for MAB in our task, we also conduct experiments with the scenario of setting utterance embedding as query and label embedding as key&value in MAB.

Based on the experiment results shown in Table 8, we can clearly observe that using label embedding as query and utterance embedding as key&value in MAB is achieving significantly higher performance than the other way around. This fact happens because the intent label is much shorter and contains very limited semantic information compared to utterances, and usually we want to set the value as the one that contains richer semantic information to fully explore the multi-head attention mechanism. Therefore, by setting query as intent label and utterance as key and values, the utterance-label embedding, the output of MAB, can directly represent how much label semantic information is contained in each corresponding utterance and PCL can further fine-grain the embedding to achieve higher classification results. However, if the query is set as utterance and the label is set as key and value, then the output will focus more on the token semantic information of utterances that associate with the limited label semantic information, where the utterance-label embedding needs to be obtained using additional pooling layer, resulting in worse utterance and label semantic association and lower classification results.

## More details on the CREDIT16 dataset

### Similar datasets.

SMS focused datasets have been collected before, for example Almeida et al. (2013) collect a dataset labeling spam messages, while Mauriello et al. (2021) label different stressors in another SMS dataset. As far as we are aware however, no dataset has been collected before that specifically deals with conversational sales interactions over SMS in the credit repair domain. In particular, conducting automated conversations over text message is a fairly new endeavor, in which we hope this dataset will be useful for benchmarking trained models.

### Initial dataset collection and domain description

- We collected an initial dataset of 115,000 unlabelled utterances acquired through SMS text message.

- The context of these conversations is an agent reaching out to a pre-subscribed customer, asking their interest in scheduling a call with a credit repair representative to discuss how their credit scores may be improved.

- As such, the utterances span discussion on customer credit scores, client interest in scheduling a call, as well as requests to remove erroneously added customers from the automated texting list.

### Pre-labelling and annotation of CREDIT16

- For the domain, we have a pre-collected training dataset with 130 intents for credit repair

| MAB | MixATIS | | MixSNIPS | | FSPS | | CREDIT16 | |
|---|---|---|---|---|---|---|---|---|
| Query/Key/Value | Acc | F1 | Acc | F1 | Acc | F1 | Acc | F1 |
| Label/Utter/Utter | **0.773** | **0.912** | **0.960** | **0.983** | **0.913** | **0.941** | **0.471** | **0.759** |
| Utter/Label/Label | 0.733 | 0.893 | 0.952 | 0.979 | 0.897 | 0.935 | 0.340 | 0.601 |

Table 8: Results of PCMID$_{base}$ pooler by swapping label embedding and utterance embedding as Query, Key and value in MAB. Label indicates label embedding and Utter indicates utterance embedding.

| Metric | Avg. # of sentences | Avg. # of tokens | Avg. # of chars | Avg. TTR | Med. Flesch reading ease |
|---|---|---|---|---|---|
| Value | 2.63 | 36.4 | 174.2 | 0.84 | 68.1 |

Table 9: Length and token type derived statistics.

and general conversational texting. We trained 130 binary text classification models, where for each classifier the negative samples for an intent were randomly collected from all other 129 intents.

- For pre-selection of which samples would be annotated by humans, we first restricted to which utterances would receive at least two labels under our automated process above. (We note that not all utterances retained more than 1 label by the end of our rounds of annotation.) Next, we split the dataset to be annotated by 3 linguists contracted to our company for a first round of annotations.

- At this point, we ran preliminary analysis and either removed or merged all labels with less than 100 examples. Merger of intents was based on discretion of which seemed common, e.g., joining together different reasons to stop calling, or merging different intents expressing interest in the service. After this label cleanup, we dropped any utterances which had no associated label.

- Next, we rotated the assignment of utterances to the linguists so that each could revise annotations from the previous round under the revised label set. The intent label distribution visualization is shown in Figure 3.

**Privacy and anonymity**

All data was run through an automatic entity classifier before even being presented to human annotators, removing all instances of PIID including name, phone number, address, email address and credit card number. The labelers then also manually reviewed given examples, removing any omissions found in the automatic redaction. All occurrences

of names are replaced by "namex" ranging from "name1" to "name2" and so forth, where the numbering indicates unique names in the utterance. Locations are replaced with "location1", "location2" and so forth. Similarly, we replace credit card info and expiration with "creditcardx" and "creditcardexpirationx", and email address with "emailaddressx".

**Basic dataset statistics**

We look at basic statistics of dataset quality. First we evaluate the average length (in characters, tokens and sentences) of each phrase. Next, we compute the average type-token-ratio(TTR) introduced by Templin 1957 which is a measure of the linguistic complexity/quality[9], and the Flesch reading ease metric introduced by Kincaid et al. 1975

The average TTR of 0.84 indicates the token repetition within phrases is limited, and a median[10] Flesch reading ease metric of 68.1 indicates the texts are easily readable by most English speakers of teenage age or older. See Table 9 for precise figures.

Next we consider the emotional context of the text using a standard T5 (Raffel et al., 2020) classifier[11] trained on the emotion dataset introduced by Saravia et al. (2018). We note that nearly 20% of examples are emotion laden, especially anger and sadness. This dataset may therefore also be useful for detection and analysis of emotion laden responses in short text conversations. See Table 10

---

[9]We calculate the TTR via the Lexical Richness package by Shen (2022)

[10]We use the *median* Flesch reading ease metric, because we find the scores are not normalized on our dataset, and has some large negative outliers that skew the distribution.

[11]See here for the emotion classifier: https://huggingface.co/mrm8488/t5-base-finetuned-emotion. We threshold any examples assigned below a 0.9 score as neutral, and with an even higher threshold of 0.97 for joy and love due to oversensitivity of the model.

| Emotion | Percentage of total | Example Utterance |
|---------|--------------------|--------------------|
| Neutral | 81.34 % | Are you all able to help me see all of my debts? |
| Anger | 8.19 % | I definitely terminated it twice now. If I find I'm still being charged somehow, I'm going to be very upset |
| Sadness | 4.61 % | I'm not well at the moment and I'm off of work on sick leave from Covid call me tomorrow evening I'm resting my voice is horrible!. |
| Fear | 0.753 % | I would love to fix my credit but I'm also afraid of spams and people stealing money. |
| Joy | 4.90 % | Thank you for checking with me. I'm in good shape right now from your service to my past issues. I have encouraged others to use your services also. |

Table 10: Emotional content of Credit-16.

for more details.

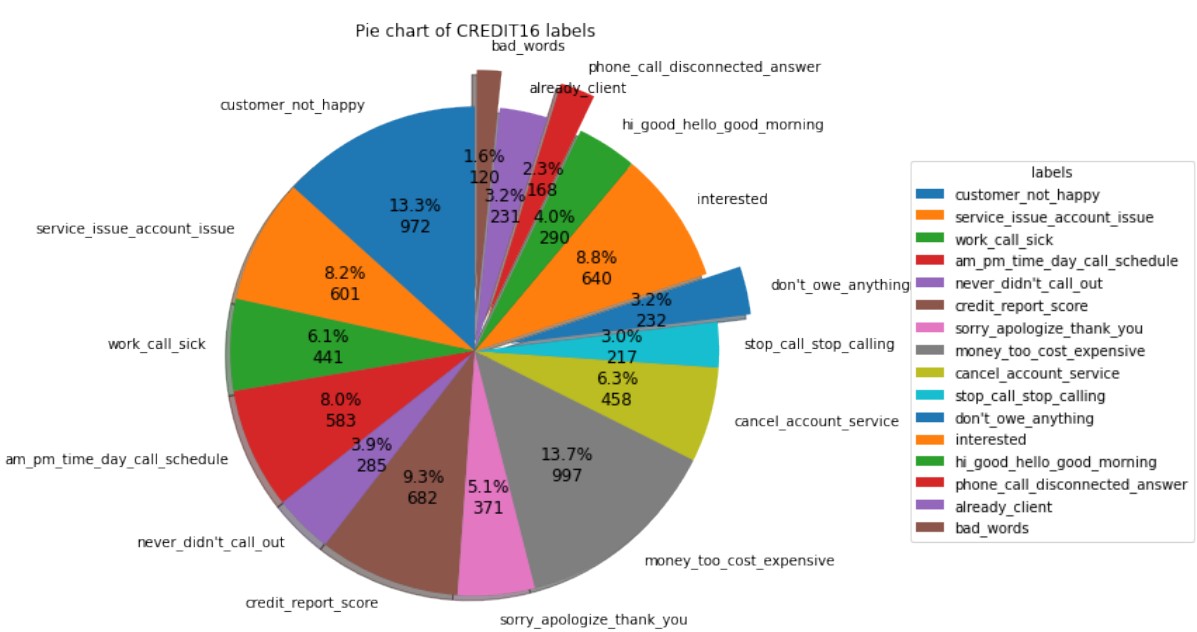

Figure 3: The distribution of CREDIT16 labels