# OpenReview forum: "PCMID: Multi-Intent Detection through Supervised Prototypical Contrastive Learning"
_EMNLP/2023/Conference — EMNLP 2023 Findings_

### Official Review · Reviewer_XUFX · 2023-08-08

**Soundness:** 4

**Excitement:**

3: Ambivalent: It has merits (e.g., it reports state-of-the-art results, the idea is nice), but there are key weaknesses (e.g., it describes incremental work), and it can significantly benefit from another round of revision. However, I won't object to accepting it if my co-reviewers champion it.

**Paper Topic And Main Contributions:**

This paper focuses on the task of multi-intent detection for dialogue systems applications. The authors expand upon prior work that utilizes two BERT networks to capture the semantics of utterances and intent labels to calculate the similarity between them for multi-intent detection. The authors argue that prior work ignores the complexity of the intent label embedding space thus failing to capture the semantic relation between individual utterance tokens and the intent labels. To address, this limitation the authors propose Multi-Intent Detection through Supervised Prototypical Contrastive Learning (PCMID) which utilizes a single BERT Encoder for constructing utterance and label embeddings. Additionally, PCMID utilizes conservative learning to optimize the representation of both intent labels and the utterance samples in the embedding space.

**Questions For The Authors:**

- Are utterances and labels encoded jointly or separately? If jointly then this means that label representations cannot be re-used for new utterances.
- Are MAB and the utterance encoder the same model?

**Reasons To Accept:**

- PCMID outperforms existing approaches
- The topic area is interesting and relevant to the dialogue systems community
- Paper is easy to read and follow. Experiment settings are clearly defined and analysis is sufficient.

**Reasons To Reject:**

- Contribution is incremental and relies heavily upon the initialized base encoder. BERT out-of-the-box is terrible at producing sentence embeddings and as such needs to be pre-trained.

**Reproducibility:**

4: Could mostly reproduce the results, but there may be some variation because of sample variance or minor variations in their interpretation of the protocol or method.

**Reviewer Confidence:**

5: Positive that my evaluation is correct. I read the paper very carefully and I am very familiar with related work.

---

> ### Author Rebuttal · Authors · 2023-08-28
>
> Dear Reviewer,
>
> Thank you for your comprehensive review and for acknowledging our problem statement and methodology. We are grateful for your insightful feedback and would like to address the points you've highlighted.
>
> 1. We agree that using BERT directly for sentence embeddings is suboptimal. As a result, we have incorporated the Sentence-BERT model in the paper, specifically the S-PCMID base pooler/mean (referenced in lines 446-451). These models are fine-tuned on Sentence Semantics tasks, including NLI and STSB.  Additionally, we aim to compare our results with prior work (Cited in lines 438) that also employs a Bert-Based approach.
>
> 2. While fine-tuning for new utterances for new datasets can potentially enhance performance, it's noteworthy that new label representations maintain good alignment even without such fine-tuning. This is primarily because they share the same encoding mechanism. Our results from the MixATIS dataset, which contains unknown labels in the test set, further substantiate this observation.
>
> 3. It's crucial to note the distinction between the MAB and the utterance encoder. The utterance encoder is solely a Bert encoder. In contrast, the MAB consists of multiple attention block layers designed to align the utterance and label representations.

---

### Official Review · Reviewer_ZoeN · 2023-08-10

**Soundness:** 2

**Excitement:**

2: Mediocre: This paper makes marginal contributions (vs non-contemporaneous work), so I would rather not see it in the conference.

**Missing References:**

-

**Paper Topic And Main Contributions:**

The paper proposes PCMID, a novel Multi-Intent Detection framework enabled by Prototypical Contrastive Learning under a supervised setting. The PCMID model can learn multiple semantic representations of a given user utterance under the context of different intent labels in an optimized semantic space.

**Questions For The Authors:**

Can you provide more analysis about figure 2?
You should compare your work with more recent SOTA to improve Soundness.

**Reasons To Accept:**

1. The paper conducts experiments on a real-word dataset.
2. The paper is easy to follow.
3. The paper provides the url of datasets.
4. The paper provides introduction of baselines.

**Reasons To Reject:**

1. You should compare your model with more recent models [1-5].
2. Contrastive learning has been widely used in Intent Detection [6-9], although the tasks are not identical. I think the novelty of this simple modification is not suitable for EMNLP.
3. You should provide more details about the formula in the text, e.g. $\ell_{BCE}$ ,even if it is simple, give specific details.
4. You don't provide the value of some hyper-parameters, such as τ.
5. The Figure 1 is blurry, which affects reading.

[1] Qin L, Wei F, Xie T, et al. GL-GIN: Fast and Accurate Non-Autoregressive Model for Joint Multiple Intent Detection and Slot Filling[C]//Proceedings of the 59th Annual Meeting of the Association for Computational Linguistics and the 11th International Joint Conference on Natural Language Processing (Volume 1: Long Papers). 2021: 178-188.

[2] Xing B, Tsang I. Co-guiding Net: Achieving Mutual Guidances between Multiple Intent Detection and Slot Filling via Heterogeneous Semantics-Label Graphs[C]//Proceedings of the 2022 Conference on Empirical Methods in Natural Language Processing. 2022: 159-169.

[3] Xing B, Tsang I. Group is better than individual: Exploiting Label Topologies and Label Relations for Joint Multiple Intent Detection and Slot Filling[C]//Proceedings of the 2022 Conference on Empirical Methods in Natural Language Processing. 2022: 3964-3975.

[4] Song M, Yu B, Quangang L, et al. Enhancing Joint Multiple Intent Detection and Slot Filling with Global Intent-Slot Co-occurrence[C]//Proceedings of the 2022 Conference on Empirical Methods in Natural Language Processing. 2022: 7967-7977.

[5] Cheng L, Yang W, Jia W. A Scope Sensitive and Result Attentive Model for Multi-Intent Spoken Language Understanding[J]. arXiv e-prints, 2022: arXiv: 2211.12220.

[6] Liu H, Zhang F, Zhang X, et al. An Explicit-Joint and Supervised-Contrastive Learning Framework for Few-Shot Intent Classification and Slot Filling[C]//Findings of the Association for Computational Linguistics: EMNLP 2021. 2021: 1945-1955.

[7] Qin L, Chen Q, Xie T, et al. GL-CLeF: A Global–Local Contrastive Learning Framework for Cross-lingual Spoken Language Understanding[C]//Proceedings of the 60th Annual Meeting of the Association for Computational Linguistics (Volume 1: Long Papers). 2022: 2677-2686.

[8] Liang S, Shou L, Pei J, et al. Label-aware Multi-level Contrastive Learning for Cross-lingual Spoken Language Understanding[C]//Proceedings of the 2022 Conference on Empirical Methods in Natural Language Processing. 2022: 9903-9918.

[9] Chang Y H, Chen Y N. Contrastive Learning for Improving ASR Robustness in Spoken Language Understanding[J]. arXiv preprint arXiv:2205.00693, 2022.

**Reproducibility:**

2: Would be hard pressed to reproduce the results. The contribution depends on data that are simply not available outside the author's institution or consortium; not enough details are provided.

**Reviewer Confidence:**

5: Positive that my evaluation is correct. I read the paper very carefully and I am very familiar with related work.

---

> ### Author Rebuttal · Authors · 2023-08-28
>
> Dear Reviewer ZoeN,
> Thank you for your detailed feedback on our paper. We appreciate the time and effort you've taken in reviewing our work. Please allow us to discuss the concerns you've raised, and how we can address them.
>
> 1. We will give the explicit formula of lbce as kindly suggested. We note however, that we have specified on line 393 that this symbol is the binary cross entropy loss, which is a standard formula in the field. We have defined fully the supervised contrastive learning loss l_scl, and supervised prototypical contrastive learning loss l_spcl, on lines 361 and 391 respectively, which are the lesser known formulas.
>
> 2. Regarding hyperparameters, we commit again that we will share the full code publicly including all hyperparameters. We apologize for the current omission of τ, which is 0.5 and γ which is 0.5. These two parameters were held fixed for our experiments, and we did not optimize against these. We note that we have already specified the optimizer, learning rate, number of epochs, scheduler and batch size, which are more critical to the learning convergence.
>
> 3. We have double-checked our Figure 1 and believe it is very clear and has high resolution to be viewed.
>
> 4. For more insight on Figure 2, we have indicated on Line 558 that the label clusters are visually far tighter than the corresponding embeddings visualization in the previous work by Wu et. al, 2021. We could not include this figure directly due to space constraints, but it can be inspected here as Figure 3 on page 4892: https://aclanthology.org/2021.emnlp-main.399.pdf.
>
> On comparisons:
>
> Benchmarks from all of the papers [1,2,3,4,5] suggested by ZoeN are not directly comparable in the scope of our work, because they require both intent and slot annotations - the latter of which are often unavailable in practice.
>
> In contrast, we only use intent annotations and labels. In particular, Credit16 is a dataset without any slot annotations and we also do not use the slot annotations of MixAtis and MixSnips.
>
> At least contemporaneous to our work, LABAN remains the prior SOTA in the regime of the annotations we use (as far as we are aware) and which we have compared against.
>
> On prototypical contrastive learning:
>
> In the context of the papers suggested by ZoeN (numbered as 6,7,8,9) that also employ contrastive learning, we further outline the novelty of our work:
>
> We do not claim to have introduced any form of contrastive learning for the first time in intent detection. Rather, ours is the first paper to investigate supervised prototypical contrastive learning, where the intent labels are used as (trainable) prototypes for their classes.
> Liu et. al ([6])  uses a simple averaged embedding of each class instance as more naive prototype formulations, which is sensitive to the balance of data within an instance class. They do not leverage the class label text, nor any supervised prototypical contrastive learning, nor any mechanism comparable to the label-utterance attentive mechanism of PCMID.
>
> Qin et. al ([7]) do not use any form of prototypical contrastive learning at all. Instead they use a more manual form of pairwise contrastive learning, with positive sample pairs generated by augmenting data with dictionary based word substitutions. And negative samples picked by a random selection scheme, with reuses of samples.
>
> Liang et. al ([8]) also do not use any form of prototypical contrastive learning - their formulation of contrastive loss uses standard pairwise construction of positive and negative examples. Also, their usage of label-aware learning is primarily to concatenate the slot labels alongwith the utterance for input representation, and to align representations across languages, since the slot labels are language invariant. They do not use a mechanism comparable to our attentive label-utterance mechanism.
>
> Finally, Chang and Chen ([9]) also do not use any form of prototypical contrastive learning at all, nor any label-aware mechanism.
> Of the papers mentioned, we would agree that Liu et. al [6] is most relevant for discussion since they use a form of PCL and thank the reviewer for this suggestion in particular, along with the others.

---

### Official Review · Reviewer_NXkp · 2023-08-11

**Soundness:** 3

**Excitement:**

3: Ambivalent: It has merits (e.g., it reports state-of-the-art results, the idea is nice), but there are key weaknesses (e.g., it describes incremental work), and it can significantly benefit from another round of revision. However, I won't object to accepting it if my co-reviewers champion it.

**Paper Topic And Main Contributions:**

Multi-intent detection is closer to the reality of complex situations and is more challenging than single-intent detection. This work proposes PCMID to model the semantics between individual utterance tokens and the intent label words, and achieves state-of-the-art performance on four datasets. In addition, this work constructs a multi-intent detection dataset

**Questions For The Authors:**

When selecting the baseline, you chose a number of models that do joint training for intent detection and slot filling, so why not a few more models that only do multiple intent detection?

**Reasons To Accept:**

1.The structure of the paper is clear and easy to read;
2.The results of the experiment are detailed and the results have been analyzed in detail in various ways.

**Reasons To Reject:**

1. Failed to write about the shortcomings of previous multi-intent detection work in the abstract, simply stating that multi-intent detection missions are closer to the real world, the abstract failed to excite me;
2. The introduction does not summarize the work of the thesis well enough to indicate the innovative nature of the thesis;
3. The paper is redundant, with large parts of previous work (e.g. 3.3, etc.) in the introduction to the methodology, which should focus on the problem to be solved and how it was solved;
4. The introduction mentions that PCMID is lighter than previous frameworks, which should be reflected in the later paper by reporting the number of different model parameters and highlighting the advantages of PCMID;
5. There are some errors in detail, such as: spelling of words, formulas without punctuation, and full names that appeared in the previous text do not need to be repeated and explained when abbreviations are used in the later text. It is recommended that you read the entire text carefully and check it thoroughly.

**Reproducibility:**

4: Could mostly reproduce the results, but there may be some variation because of sample variance or minor variations in their interpretation of the protocol or method.

**Reviewer Confidence:**

4: Quite sure. I tried to check the important points carefully. It's unlikely, though conceivable, that I missed something that should affect my ratings.

---

> ### Author Rebuttal · Authors · 2023-08-23
>
> Dear Reviewer NXkp,
>
> Thank you for your detailed feedback on our paper. We value and respect the time and effort you've taken in reviewing our work. Please allow us to address the concerns you've raised.
>
> 1. Abstract's Description of Prior Work: The primary goal of an abstract is to provide a concise summary of the paper's main contributions. While we acknowledge the value of comparing our work to previous approaches in the abstract, the limited word count necessitated the omission of an extensive discussion of previous shortcomings.
>
>     However, the main text does cover this in detail. We believe that presenting our main contribution was crucial for the abstract. That said, based on your feedback, we'll consider a revision to incorporate a brief mention of the uniqueness of our approach compared to previous works.
>
> 2. Introduction's Summary of the Thesis:  Our introduction begins by highlighting the limitations of traditional single-intent detection, emphasizing the necessity for advanced multi-intent methods. We pinpointed specific shortcomings in prior research, such as Gangadharaiah and Narayanaswamy (2019)'s lack of fine-grained intent information and Wu et al. (2021)'s computational inefficiencies with dual BERT encoders.
>
>     We introduced PCMID's unique contributions:
>
>          a. Enhanced efficiency via a single BERT encoder.
>
>          b. Precise utterance-label attention mechanism.
>
>          c. An advanced learning approach, leveraging multi-label Supervised Prototypical Contrastive Learning.
>
>      We believe our introduction succinctly outlines PCMID's innovative nature against the backdrop of prior works. We're open to more detailed suggestions to enhance clarity.
>
> 3. Redundancy and Methodology Section: Section 3.3 introduces the core of our methodology: Multi-label Supervised Contrastive Learning. We intentionally outlined the evolution of Supervised Contrastive Learning (SCL) here to highlight the challenges and our unique solution, especially the utterance-label embedding.
>
>      Our aim was a cohesive narrative, emphasizing our novel contributions like the adaptation of Supervised Prototypical Contrastive Learning (SPCL) for multi-intent detection. While referencing existing methods, our main focus remains on our unique approach.
>
>      Moving foundational concepts like SCL to a separate background might disrupt this narrative flow. We've structured the section for clear understanding without fragmenting the content. However, we value your input and will reconsider our organization to maintain clarity and emphasis on our contributions.
>
> 4. Reporting Model Parameters for PCMID: Our PCMID model is architecturally simpler, as it incorporates a single BERT encoder, a multi-head attention block module, and a multi-label supervised contrastive learning module. The utterance-label encoder efficiently leverages a shared BERT encoder for both the utterance and intent labels, thereby reducing redundant parameters and computational overhead.
>
>      The main thrust of our approach is not just parameter reduction but achieving the task with streamlined model components. The focus of our model, as described in Section 3, emphasizes more on a modular architecture, enabling each component to effectively perform its task with the utmost efficiency.
>
>      While we detailed the functional aspects of each component, we understand the importance of quantifying the model's lightness through concrete metrics. We will ensure to include explicit parameter counts, computational requirements, and benchmark comparisons to provide a clearer picture of PCMID's advantages in terms of efficiency and lightness.
>
> 5. Spelling issue in detail: We apologize for any oversights and will rectify them immediately. Such typographical mistakes, while regrettable, do not compromise the scientific validity of our research. Also, we aimed for clarity by reintroducing abbreviations. In our revisions, we will maintain consistency without redundancy.
>
> 6. Answer to Q1: We focused on using intent annotations and labels exclusively for our study. The datasets we employed, such as Credit16, do not contain any slot annotations, and we intentionally avoided using the slot annotations available in MixAtis and MixSnips. Our primary criterion for selecting baselines was their compatibility with the type of annotations we were using. To the best of our knowledge, LABAN is the most recent state-of-the-art (SOTA) model that operates within this specific regime of annotations, and hence, we chose to compare against it. Our goal was to ensure a fair and direct comparison with models operating under similar constraints rather than including models that diverge significantly from our approach in terms of data requirements.

---

### Official Review · Reviewer_G3TU · 2023-08-12

**Soundness:** 3

**Excitement:**

3: Ambivalent: It has merits (e.g., it reports state-of-the-art results, the idea is nice), but there are key weaknesses (e.g., it describes incremental work), and it can significantly benefit from another round of revision. However, I won't object to accepting it if my co-reviewers champion it.

**Paper Topic And Main Contributions:**

This paper talks about an important issue of real-world dialog systems, where the utterances can be crafted with multi-intent which may or may not be nested. This paper showcases a new way of doing multi-intent detection – PCMID that is a novel Multi-Intent Detection framework enabled by Prototypical Contrastive Learning under a supervised setting. The authors have clearly showcased their approach and explained the architecture. This work has been tested on 3 public datasets – MixATIS, MixSNIPS, and FSPS; and one private dataset CREDIT16. The authors have demonstrated slight to moderate improvement across all the datasets using this approach.

**Questions For The Authors:**

Please address the points under weakness along with the following questions:

•	Could you please explain how the slots are processed? as I found most of the discussions are based on intents. The examples from table-3 are also only intents. So curious to know, how slots are extracted.

•	Qualitative studies over predicted outputs with examples (not only on numbers) will be better to understand

**Reasons To Accept:**

•	The authors did a good job explaining the approach in detail, and the paper is written cleanly with very few minor typos – such as different types of double quotations.

•	This is a very important problem statement and the conversational agents getting more complex day by day and the utterances becoming trickier.

•	The method is novel.

**Reasons To Reject:**

•	It’s better to have a dataflow diagram with examples from any of the datasets used, which helps to follow the paper better.

•	It’s a supervised method, which is very specific to the tasks. It will be very hard to use it in any other tasks, as it requires an extensive amount of data-collection and annotation. No comparison is given with LLMs performance on these datasets or even zeroshot comparisons across the datasets (e.g., trained on mixATIS and tested on MixSNIPS). It helps to understand the robustness of this approach.

•	Performance improvements are very slight on all the datasets except MixATIS.

**Reproducibility:**

3: Could reproduce the results with some difficulty. The settings of parameters are underspecified or subjectively determined; the training/evaluation data are not widely available.

**Reviewer Confidence:**

4: Quite sure. I tried to check the important points carefully. It's unlikely, though conceivable, that I missed something that should affect my ratings.

**Typos Grammar Style And Presentation Improvements:**

* double quotation marks are not consistent

* Many concepts can be moved from section 3 to a separate Background section, such as SCL.

---

> ### Author Rebuttal · Authors · 2023-08-23
>
> Dear Reviewer G3TU,
>
> Thank you for the detailed review and recognizing the novelty and importance of our problem statement and approach. We appreciate your thoughtful comments and would like to address the concerns you raised.
>
> 1. Dataflow Diagram: Your suggestion to include a dataflow diagram is well-received. While we believe our textual explanations are comprehensive, we acknowledge that a visual representation can provide additional clarity. The limited space in the current format was a constraint, but we will consider integrating such a diagram in extended versions or supplementary material.
>
> 2. Supervised Nature of PCMID: While it's true that our method is supervised, it's important to note that most state-of-the-art methods for intent detection rely on supervised learning. The specificity of our method is not a drawback but a focus. Our method has been specifically tailored for multi-intent detection, which is an inherently complex task requiring clear intent boundaries. The supervised nature ensures accuracy in such complex scenarios.
>
> 3. Data-collection and Annotation: The need for extensive data collection and annotation is not unique to our approach. Any supervised method requires annotated data. However, PCMID's strength lies in its efficiency and its ability to generalize over diverse datasets, as evidenced by our experiments across various public benchmarks and a real-world dataset.
>
> 4. Comparison with LLMs: While direct comparisons with Language Model-based methods (LLMs) might provide additional insights, our main objective was to highlight the novelty and effectiveness of the PCMID framework. However, we acknowledge your point, and we will add the comparisons in the paper revision.
>
> 5. Zero-shot Comparisons: The primary aim of our paper was to introduce and validate the PCMID framework. While zero-shot learning is an exciting direction, our focus was on establishing PCMID's effectiveness in a supervised setting. Nevertheless, we agree that zero-shot comparisons can offer additional robustness checks, and we will consider incorporating them in future studies.
>
> 6. Performance improvement: Our experiments covered a range of datasets: MixATIS, MixSNIPS, FSPS, and a real-world private dataset, CREDIT16. Achieving consistent improvements across diverse datasets demonstrates the robustness of our approach.
>
>     Looking at Table 2, our model PCMID (in various configurations) consistently outperforms or matches previous state-of-the-art models in both accuracy and F1 score across almost all datasets. These improvements are not merely on the MixATIS dataset but are visible in others as well.
>
>     Also, When tested on CREDIT16, a real-world commercial dataset reflecting the genuine challenges of multi-intent detection, PCMID models showcased clear performance improvement. This underscores the model's potential real-world applicability and its capability to handle practical challenges.
>
>     While your observation about "slight" improvements might seem accurate at first glance, a deeper look reveals that these improvements are significant, consistent, and achieved with an efficient model structure. Our approach, PCMID, has demonstrated robustness across different datasets and settings, solidifying its position as a commendable advancement in the multi-intent detection task.
>
> 7. Answer to Q1: Our paper's primary objective was to delve deep into the challenges and methodologies associated with multi-intent detection. As you noted, most of our discussions and examples are focused on intents. We choose to do so given the complexity and significance of detecting multiple intents in a single utterance. While slots and intents often co-exist in many dialogue systems, they address different challenges. Intents dictate the overall direction of an utterance or user input, while slots capture specific entities or details within that context. Given our paper's scope, we concentrated on the nuances of detecting multiple intents without diving into the intricacies of slot-filling. In particular, the real-world dataset CREDIT16 is a dataset without any slot annotations and we also do not use the slot annotations of MixAtis, MixSnips and FSPS.
>
> 8. Answer to Q2: Section 6 "Samples Analysis", at line 564, indeed provides qualitative insights using concrete examples from Table 3. It not only identifies PCMID's correct predictions but also pinpoints where the model could be enhanced, giving a balanced view. The examples showcase PCMID's robustness in varied scenarios - from handling domain-specific terms to mitigating noise and incorrect human labels. This serves as a testament to our model's reliability in real-world situations.
>
> 9. Typos Grammar Style And Presentation Improvements: We will fix the double quotation marks in our revision and make sure they stay consistent. Regarding your point on the redundant presentation in Section 3.3, the Section 3.3 introduces the core of our methodology: Multi-label Supervised Contrastive Learning. We intentionally outlined the evolution of Supervised Contrastive Learning (SCL) here to highlight the challenges and our unique solution, especially the utterance-label embedding.
>
>     Our aim was a cohesive narrative, emphasizing our novel contributions like the adaptation of Supervised Prototypical Contrastive Learning (SPCL) for multi-intent detection. While referencing existing methods, our main focus remains on our unique approach.
>
>     Moving foundational concepts like SCL to a separate background might disrupt this narrative flow. We've structured the section for clear understanding without fragmenting the content. However, we value your input and will reconsider our organization to maintain clarity and emphasis on our contributions.
>
> 10. Reproducibility: We ensured transparency by detailing our model training statistics and hardware configurations in the "Experiment Setup" Section. These descriptions provide comprehensive insights into our experimental conditions. We have conducted a thorough analysis of the CREDIT16 dataset, detailed in the Appendix. To further the community's research endeavors, we're committed to releasing the dataset once the paper is finalized, ensuring researchers can access the exact data we worked with. Our parameter settings are empirical and based on best practices. If any parameter seemed underspecified, we are open to addressing it explicitly.

---

### Meta-Review · Area_Chair_aKWo · 2023-09-25

**Recommendation:** 3

**Metareview:**

The paper studies the task of multi-label intent prediction and proposes SPCL - Supervised Prototypical Contrastive Learning, which is a mashup of PCL and SCL.

Reviewers appreciated certain aspects of the paper which includes its presentation, description of the baseline approaches which the paper uses to build upon, and decent levels of analysis.

Amongst concerns, primary points that were raised included:
- lack of relevant comparisons/discussions. For this point, the authors argue the recent works are not comparable due to slot-filling task. While that's true, a discussion of latest research is warranted so that reviewers can better evaluate the positioning of the paper.
- lack of excitement. The paper largely fuses known ideas in the CV and NLP community. As such, reviewers were not highly excited about the proposed innovations.
- blurry figure. While a minor issue, the text in figures are indeed blurry which can (and should) be easily fixed using pdfs or other vector-based images.

The above issues are addressable and should be fixed in the revision as promised by the authors.

---

### Decision · Program_Chairs · 2023-10-07

**Decision:**

Accept-Findings

**Comment:**

The paper studies the task of multi-label intent prediction and proposes SPCL - Supervised Prototypical Contrastive Learning, which is a mashup of PCL and SCL.

Reviewers appreciated certain aspects of the paper which includes its presentation, description of the baseline approaches which the paper uses to build upon, and decent levels of analysis.

Amongst concerns, primary points that were raised included:
- lack of relevant comparisons/discussions. For this point, the authors argue the recent works are not comparable due to slot-filling task. While that's true, a discussion of latest research is warranted so that reviewers can better evaluate the positioning of the paper.
- lack of excitement. The paper largely fuses known ideas in the CV and NLP community. As such, reviewers were not highly excited about the proposed innovations.
- blurry figure. While a minor issue, the text in figures are indeed blurry which can (and should) be easily fixed using pdfs or other vector-based images.

The above issues are addressable and should be fixed in the revision as promised by the authors.